# Flushing Efficiency of Run-of-River Hydropower Plants: Novel Approaches Based on Physical Laboratory Experiments

**Thomas Gold *** , **Kevin Reiterer** , **Christoph Hauer, Helmut Habersack** and **Christine Sindelar**

CD-Laboratory for Sediment Research and Management, Institute of Hydraulic Engineering and River Research, Department of Water, Atmosphere and Environment (WAU), University of Natural Resources and Life Sciences, Am Brigittenauer Sporn 3, 1200 Vienna, Austria; kevin.reiterer@boku.ac.at (K.R.); christoph.hauer@boku.ac.at (C.H.); helmut.habersack@boku.ac.at (H.H.); christine.sindelar@boku.ac.at (C.S.)
* Correspondence: thomas.gold@boku.ac.at

**Abstract:** Periodic flushing operations during moderate flood events ($\leq$annual flood flow $HQ_1$) are an approach to counteract problems caused by disturbed sediment continuity in rivers, which is possibly an effect of run-of-river hydropower plants (RoR-HPPs). Considering ecology, flood risk, technical, and economical reasons, discharge values of $0.7 \times HQ_1$ are a good reference point for the initiation of gate operations. This work aimed to investigate the role of different gate opening actions on the effectiveness of such flushing measures. Physical model tests were performed, to capture bed load rates, together with 2D velocity measurements in the vicinity of two movable radial gates above a fixed weir. The length scale of the idealized model arrangement was 1:20, and a conveyor-belt sediment feeder was used to supply a heterogeneous sediment mixture. Velocities were acquired using 2D laser doppler velocimetry (LDV). Based on the LDV measurements, mean velocity profiles and Reynolds stresses were derived. The full opening of both radial gates led to the highest bed load mobility. While the flushing efficiency drastically decreased, even for slightly submerged gates, an asymmetrical gate opening initially led to the formation of a flushing cone in the vicinity of the weir, accompanied by temporarily high flushing efficiency. In conclusion, our results stress the importance of full drawdowns in successfully routing incoming bed load downstream of the HPP. However, the combination of an asymmetric gate opening followed by a full drawdown could be a promising approach to further improve the flushing efficiency of RoR-HPPs.

**Keywords:** sediment management; run-of-river hydropower; gate positions; physical modeling; laser doppler velocimetry; Reynolds stress; Shields curve

## 1. Introduction

A run-of-river hydropower plant (RoR-HPP) is a type of hydroelectric generation plant that uses transversal structures and turbines to harvest the kinetic energy carried by water, in order to generate electricity. RoR-HPPs typically provide little-to-no storage capacity. However, decreasing flow velocities and backwater effects accompanied by impounding can lead to the trapping of incoming bed load, potentially disturbing the sediment continuity of rivers worldwide [1–3]. During the projected operations of a hydropower facility, nearly 100% of the incoming bed load material is held back in the impoundment [4]. The disrupted sediment dynamics can impact the downstream river reach, potentially leading to bed erosion and land loss [5].

A previous experimental study by Sindelar et al. [6] investigated the effects on sediment continuity of weir height and reservoir widening at RoR-HPPs in gravel-bed rivers. Their findings suggested new concepts of low weir heights and cross-sectional reservoir widths in the design of low-head RoR-HPPs, to facilitate frequent and efficient flushing operations. This is expected to enhance sediment continuity, and to reduce maintenance and operational costs. Sustainable sediment management strategies, including the removal

of sediment deposits by flushing, will potentially extend the life expectancy of the RoR impoundments [7,8].

The research of Sindelar et al. [2] dealt with delta formation at operation level at the reservoir head of RoR-HPPs in gravel-bed rivers. Their findings showed that delta formation has the potential to increase the risk of high floods, and they suggested the initiation of reservoir drawdown for flow rates of $0.7 \times HQ_1$. Reiterer et al. [3] investigated the consequences of drawdown flushing operations on delta formations at the headwater section of an RoR-HPP, for flow rates of $0.7 \times HQ_1$. Retrogressive erosion processes were found to be dominant during the initial flushing phase, accompanied by high sediment transport rates, proving the effectiveness of low-flood-flow flushing operations (LFFFO). If the drawdown is partial, the bed load material is expected to resettle downstream in the impoundment [3]. Gold and Reiterer [9] performed experiments gauging the flushing effects in the vicinity of an RoR-HPP. When the radial gates were only slightly submerged, the flushing efficiency dropped to insufficient values.

Different gate positions significantly affect the flow conditions in the vicinity of a weir. In addition, bed forms and the mobility of bed load correlate with the shape and magnitude of flow velocity and shear stress, and not only in the boundary layer of the bed [10,11]: Hanmaiahgari et al. [10] also described a decrease in the velocity gradient $du/dy$ with the increasing mobility of the sediment bed. This decrease in velocity near the bed was explained by the extraction of energy from the mean flow, which was transferred to the bed particles, to sustain bed mobility [10]. Furthermore, they found a strong influence on the normalized velocity distribution near the bed for mobile bedforms, while no significant change with increasing flow rate (and Reynolds number) for immobile beds and incipient sediment motion was present. The von Kármán constant $\kappa$ is commonly used to describe the relation between flow velocity and friction velocity. Earlier research [12,13] showed a decrease of $\kappa$ with increasing bed movement. Hanmaiahgari et al. [10] described a decrease of $\kappa$ with increasing bed form motion, due to the increased thickness of the roughness sublayer. For incipient bed motion, weak bed forms, and mobile bed forms (strong bed motion), the classical logarithmic law and profile is therefore not applicable [10].

The aim of the present study was to further investigate the effect of gate opening positions on flushing efficiency during LFFFO ($0.7 \times HQ_1$). Our model resembled a small-to-medium-sized alpine-gravel-bed river. An idealized model setup of two symmetrical radial gates placed on top of a low-height weir, with a length scale of 1:20, was used for the experiments. The physical model contained a mobile bed with a heterogeneous sediment mixture and a constant sediment supply. Three different scenarios—two with symmetrical, and one with asymmetrical gate positions—are presented. For the asymmetrical gate arrangement, the hydro-morphological effects are discussed, based on a digital elevation model (DEM). By comparing the flushing efficiency of the different scenarios, we present suggestions for more sustainable flushing operations. By conducting 2D laser doppler velocimetry measurements (2D LDV), we analyzed the effects of the gate position on the hydrodynamic properties of the flow. The findings are discussed, based on the Shields diagram [14], velocity, Reynolds stress profiles, and the existing literature.

## 2. Materials and Methods

### 2.1. Sediment Flume and Model Scaling

The experiments were conducted in a hydraulic glass flume, which contained three sections: (i) an inlet basin; (ii) the experimental section of 10 m length and 1 m width; and (iii) an outlet section with an automated gate, to control water levels and flow conditions. The water was recirculated back to the inlet section by a frequency-controlled pump at the outlet section. Figure 1 illustrates the experimental setup. The bed surface in the experimental section comprised a heterogeneous unicolor sediment mixture, ranging from 0.0007 to 0.006 m, representing a gravel riverbed with grain size fractions ranging from 0.014 to 0.12 m. At the beginning of the flume, a programmable sediment feeder (SF) supplied the sediments. Furthermore, the transported sediments were collected in a sediment trap

(ST) which was located near the outlet section. The weir model (WM), with two adjustable radial gates, was located at the end of the experimental section. The weir model had a total width of 1 m, and each movable gate section was 0.35 m wide. The maximum water depth, with gates closed and resting water, was 0.26 m above the fixed weir crest of 0.05 m height. A section of the WM, with three different gate positions, is depicted in Figure 3. A camera system (PHOTO) with a Nikon D7100, a linear positioning system, and point markers along the flume, enabled the derivation of a digital elevation model (DEM) by means of photogrammetry. Therefore, the recordings were processed by Agisoft Metashape 2.0.1. Regarding the laser doppler velocimetry system, a detailed explanation can be found in Section 2.2.

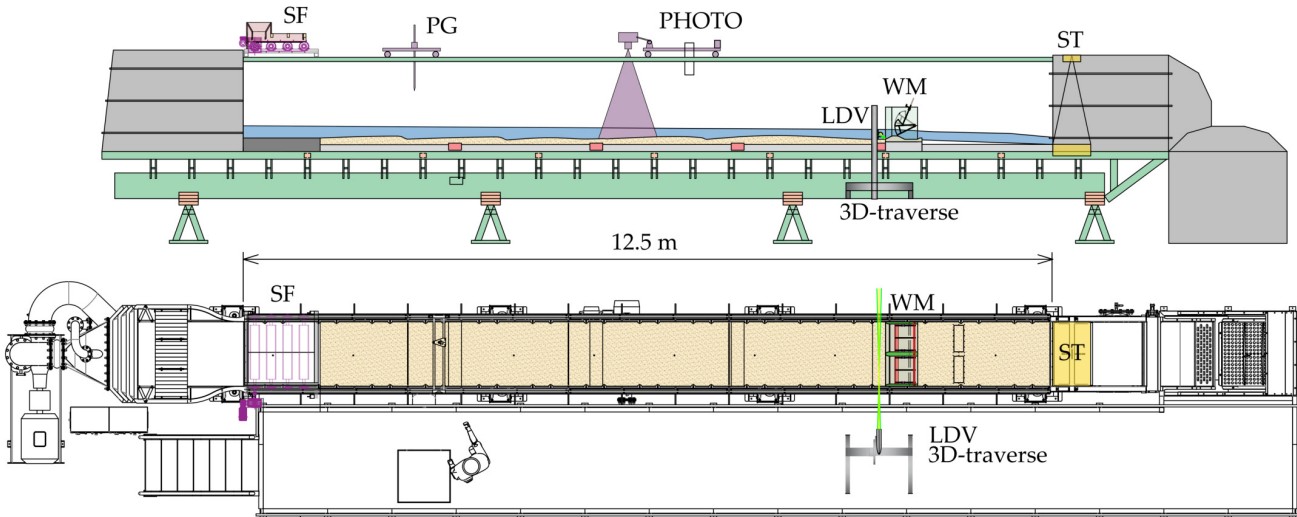

**Figure 1.** Experimental flume and measurement equipment: top = longitudinal section; bottom = plane view; LDV = laser doppler velocimetry; PG = point gauge; PHOTO = camera system for photogrammetry; SF = sediment feeder; ST = sediment trap; WM = weir model. Figure not to scale.

The investigated model setup represented a small-to-medium-sized alpine-gravel-bed river with an idealized model HPP containing two symmetrical radial gates placed on top of a low-height weir. The physical model was scaled down by use of the model law of Froude similarity, which is commonly used for free surface flows, weir hydraulics and hydropeaking. Froude similarity is provided when the Froude number in the model matched that in the prototype [15]:

$$Fr_r = \frac{v_r}{\sqrt{g_r L_r}} = 1, \qquad (1)$$

where $Fr_r$ is the ratio of the Froude numbers in the prototype and the model, $v_r$ is the velocity scale, $L_r$ is the geometrical length scale, and $g_r$ is the ratio between the gravitational forces. By rearranging Equation (1), the velocity scale $v_r$ is obtained. As $g_r$ is usually 1, $v_r$ is derived as follows:

$$v_r = \sqrt{g_r L_r} = \sqrt{L_r}. \qquad (2)$$

With the given geometrical length scale, $L_r$ = 1:20, all other model quantities could be derived, based on the law of Froude similarity [15]. Table 1 gives a summary of the scaling relations and the corresponding downscaled model quantities.

**Table 1.** Summary of model scaling based on the law of Froude similarity.

| $L_r\,^1$ | $A_r\,^1$ | $v_r\,^1$ | $t_r\,^1$ | $Q_r\,^1$ |
|---|---|---|---|---|
| $L_n/L_m$ | $L_r^2$ | $L_r^{0.5}$ | $L_r/v_r = L_r^{0.5}$ | $v_r \cdot A_r = L_r^{(5/2)}$ |
| 1:20 | $(1:20)^2$ | $(1:20)^{0.5}$ | $(1:20)^{0.5}$ | $(1:20)^{5/2}$ |

$^1$ $L_r$ is the geometrical length scale, $A_r$ is the areal scale, $v_r$ is the velocity scale, $t_r$ is the timescale, and $Q_r$ gives the scaling of the flow rate.

For an undistorted model with length scale $L_r$ = 1:20 and Froude similarity, the hydraulic timescale $t_r = \sqrt{L_r}$. Hence, two hours of experiment referred to $\approx$9 h in nature. Considering the maximum grain diameter ($d_s$) as the characteristic length scale of the bed roughness height and the kinematic viscosity $\nu = 10^{-6}$ m$^2$s$^{-1}$, the grain Reynolds number $Re^* = u^*/(d_s\nu)$ was >100 for each scenario: This justified the assumption of the sediment timescale $t_s = t_r$ [2]. A more comprehensive explanation regarding the scaling of sediment transport can be found in [2,15].

*2.2. Laser Doppler Velocimetry (LDV)*

Laser doppler velocimetry is a well-proven, non-contact, non-invasive method for measuring flow velocities [16]. It works by means of the intersection of two laser beams at an intersection point for a single velocity component. For the present study, 2D LDV measurements, in natural turbid water in the absence of artificial seedings, were conducted. In the case of two pairs of laser beams, the technique is referred to as 0D2C (0 dimensions and 2 velocity components). We measured the longitudinal ($u$) and vertical ($v$) velocity components along a vertical profile in the weir field axis.

The present system is the FlowExplorer from DANTEC Dynamics, which consists of a calibrated laser probe mounted on a traverse system. This made it possible to automatically record all measuring points of a vertical profile of the cross section. The reflected signal was evaluated by a burst spectrum analyzer, and the output was analyzed using the supplied software (BAS Flow v5). Figure 2 shows images of the LDV setup for the experiments. The acquired velocity measurements were used to compute mean velocity profiles for each scenario. Based on Reynolds decomposition of the velocity measurements, the Reynolds stress was calculated:

$$u = \overline{u} + u' \text{ and } v = \overline{v} + v', \tag{3}$$

where $u$, $v$ are the measured velocities, $\overline{u}$, $\overline{v}$ are the time averages for the measurement point, and $u'$, $v'$ denote the turbulent velocity fluctuations. The Reynolds stress for a single measurement point writes as:

$$R_{ij} = -\overline{u'v'}. \tag{4}$$

Normalization was performed, based on the square of the shear velocity $u^{*2}$:

$$u^* = \sqrt{\frac{\tau_0}{\rho}}, \tag{5}$$

where $\rho$ is the fluid density and $\tau_0$ is the reference shear stress.

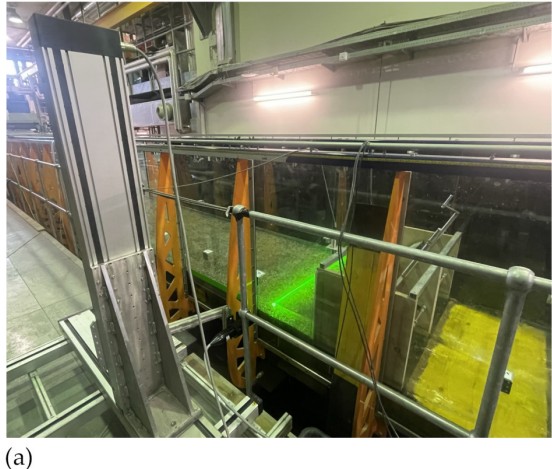
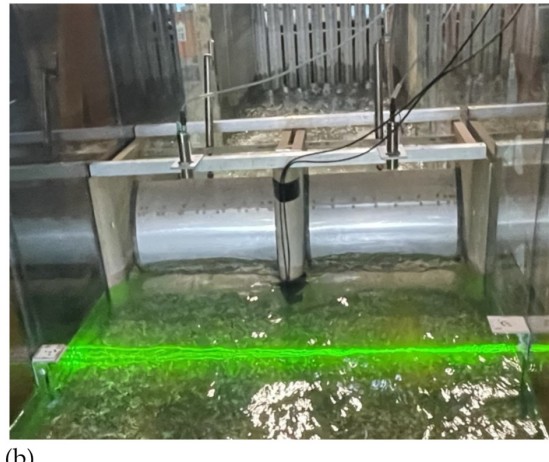

(a)                                                                                      (b)

**Figure 2.** Experimental setup for LDV measurement through a glass wall of the flume: (**a**) automated 3D positioning system (traverse) with mounted LDV probe; (**b**) LDV measurement with both radial gates slightly submerged.

### 2.3. Experimental Procedure and Conditions

As a first step, suitable initial conditions for the experiments had to be accomplished. Therefore, the experimental section within the flume was filled with a heterogeneous sediment mixture (mean diameter $d_m$ of 0.0024 m), and it was scraped evenly with slope I = 0.0035. Next, the equilibrium transport rate without the weir model was determined iteratively. The fed sediment input $SI_{ti}$ was equated to the measured output $SO_{ti-1}$, using an iteration time step of 30 min. This process was pursued, until the mean bed slope S matched I, and $SI_{ti}$ converged towards $SO_{ti}$.

All the experiments were conducted at a steady flow rate of 0.07 $m^3s^{-1}$. Subsequently, during the actual experiments, a constant amount of sediment, at the rate of the equilibrium transport (SIC), was fed. The experiments were conducted under steady flow conditions provided by the frequency-controlled pump. During the test run, the LDV measurements were performed 0.3 m upstream of the weir model. The recording length for each measurement point was 30 s. Depending on the present water depth, up to 25 points per profile were measured. Every 15 min or 30 min, the test run was carefully interrupted, by closing the automated gate at the outlet section and simultaneously decreasing the flow rate to zero. During this phase, the bed load collected in the sediment trap was weighted, and the trap was emptied again. After the model was carefully drained, image recording of the bed surface was conducted, to derive the required DEMs by means of photogrammetry.

In this study, three scenarios (S1, S2, S3), with different gate opening positions, were investigated. Scenarios S1 and S2 were divided into four sequences of 30 min each, amounting to a total time of two hours. For scenario S3, firstly, two sequences of 15 min, and secondly, three sequences of 30 min, were recorded. Table 2 sums up the experimental conditions. Furthermore, the gate positions and flow conditions for S1, S2, and S3 are depicted in Figure 3, which, for S3, shows the closed gate, while the open gate is not visible.

**Table 2.** Summary of experimental conditions for the three scenarios (S1, S2, S3).

| Scenario | Gate Position | Duration (s) | SIC [1] (kgh$^{-1}$) | Q (m$^3$s$^{-1}$) |
|---|---|---|---|---|
| S1 | Both open | 4 × 1800 | 60 | 0.07 |
| S2 | Both slightly submerged | 4 × 1800 | 60 | 0.07 |
| S3 | One closed, one open | 2 × 900 and 3 × 1800 | 60 | 0.07 |

[1] SIC denotes the constant sediment input, and Q is the flow rate.

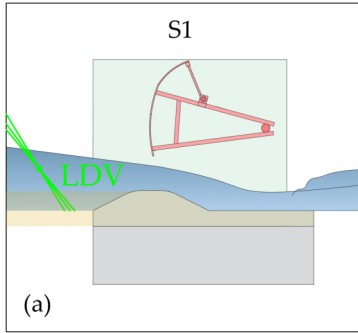
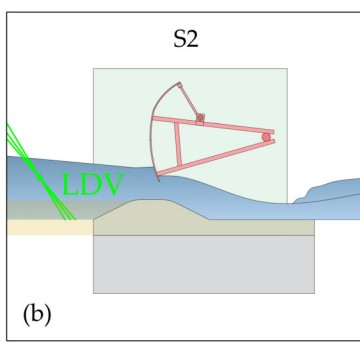
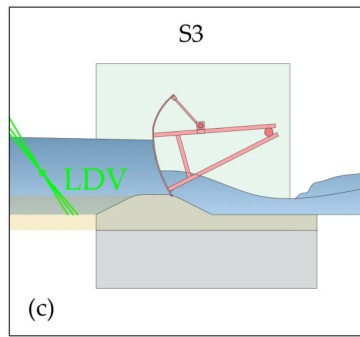

**Figure 3.** Gate opening patterns during the experiments: (**a**) both radial gates fully open; (**b**) both radial gates submerged (20% of the water depth from S1); (**c**) one radial gate closed, one fully open.

## 3. Results

### 3.1. Flushing Efficiency and Morphological Effects

For the present study, the measure for the efficiency of the flushing operation was defined as the ratio between sediment output and sediment input, as depicted in Figure 4.

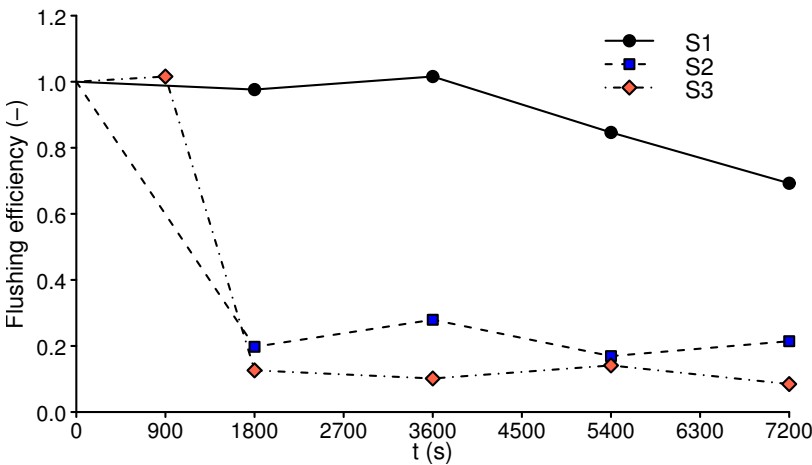

**Figure 4.** Comparison of the flushing efficiency (sediment output to input ratio) for the observed scenarios. Every measurement point represents the sediment output to input ratio for a sequence of 900 or 1800 s.

With both radial gates fully open, the highest efficiency was observed. During this scenario, S1, the overall flushing efficiency, was about 0.9. The decline in the flushing efficiency for S1 over time is explained by the natural range of fluctuation of the bed load transport and by bed form alteration (dunes). During several hours of preliminary test runs at the same flow rate (LDV calibration), bed load fluctuations with the same order of magnitude could be observed. When the gates where only slightly submerged (20% of the water depth from S1), the flushing efficiency decreased to 0.22, which indicates the importance of free flowing conditions. The drop in sediment mobility, with the commencement of the submergence of the gates, confirms the findings of Gold and Reiterer [9]. If the deposited bed load material should be routed through the full length of the impoundment, partial drawdowns will be insufficient, as suggested by Reiterer et al. [3]. One radial gate fully open and the second one fully closed led to an initially high flushing efficiency, which rapidly decreased to values around 0.10.

The initially high values of S3 were related to a 3D erosion process in the vicinity of the weir. A cone-shaped scouring structure occurred, as depicted in Figure 5. The formation of

the flushing cone was explained by the presence of significant cross flow in the vicinity of the weir. The erosion depth was limited to the thickness of the experimental sediment layer, and the width of the scouring amounted to about 0.7 m. The total volume eroded during the scouring process amounted to about 0.010 m$^3$. Aside from this local erosion cone, only a little mobility of the mobile bed (incipient-like motion) was observed for S3. After about 900 s, the local scouring process mitigated, and the flushing efficiency dropped to 0.10. Table 3 provides an overview of the relevant measurement results. For scenario S1, no significant changes in bed morphology were present, and the bed load was transported in the form of dunes (strong bed form mobility). For S2, only weak bed mobility was observed without significant bed form structures.

**Table 3.** Summary of experimental results (mean values) for the three scenarios (S1, S2, S3).

| Scenario | $h$ [1] (m) | $u_{max}$ [1] (ms$^{-1}$) | $\tau_0$ [1] (Nm$^{-2}$) | Sediment Output [2] (kgh$^{-1}$) | Flushing Efficiency |
|---|---|---|---|---|---|
| S1 | 0.105 | 0.78 | 3.8 | 54 | 0.90 |
| S2 | 0.130 | 0.57 | 1.8 | 13 | 0.22 |
| S3 | 0.165 | 0.56 | 0.8 | 12 | 0.20 |

[1] $h$ is the water depth, $u_{max}$ is the maximum velocity, and $\tau_0$ is the reference shear stress (bed shear stress).
[2] Average over time.

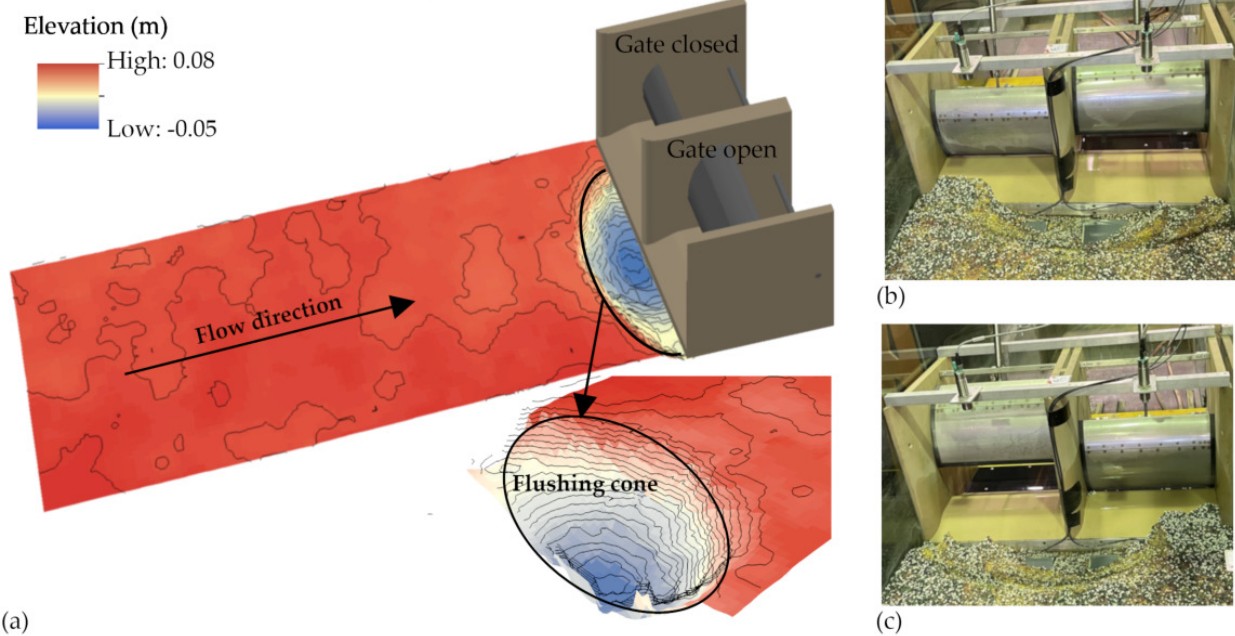

**Figure 5.** Flushing cone of scenario S3: (**a**) digital elevation model (DEM) with the weir structure; (**b**) photograph after experimental run S3, showing the flushing cone; (**c**) photograph after validation experiment for scenario S3, with inverted gate opening arrangement showing a mirrored shaped erosion pattern.

### 3.2. Velocity Profiles and Reynolds Stress from LDV

The effects of the gate position on the flow dynamics were investigated by analyzing the LDV measurements. The measured maximum velocities $u_{max}$ and bed shear stresses $\tau_0$ in all three scenarios are provided in Table 3. Already, a slight submergence of both weir gates (S2) had led to a halving of the measured bed shear stresses and to a decrease in the maximum velocities, of almost 30%. This sharp decline in both the velocities and the bed shear stresses was also directly reflected in, respectively, the measured sediment yield and the flushing efficiency. In Figure 6, time-averaged velocity profiles of the three scenarios are displayed. As the scenarios had different water depths and velocity magnitudes, a

normalization was performed. While in the upper part only small differences between the scenarios were present, large differences in the shape of the profile appeared in the near bed region. According to Faruque and Balachandar [17], a proper outer velocity scale in turbulent boundary layers should be the freestream velocity $u_{max}$. Accordingly, we performed normalization with the direct measured quantities $u_{max}$ for the velocity and the water depth $h$ for the y-coordinate. Our results showed a significant influence of the sediment mobility on the normalized velocity distribution. With increasing sediment mobility, the velocity gradient decreased. Additionally, in Figure 6a, two velocity profiles from Hanmaiahgari et al. [10] (SM2), incipient sediment motion (IN), and one velocity profile with mobile bed forms (SM2), are plotted.

The present results show a downward shift from the smooth-wall log law with increasing bed motion, as indicated by the black arrow in Figure 6a. From the velocity measurements, Reynolds stress profiles were derived, using Reynolds decomposition. In Figure 6b, normalized Reynolds stress profiles are plotted, together with data from Hanmaiahgari et al. [10]. Again, the length scale for the y coordinate was $h$, and $-\overline{u'v'}$ was normalized using the square of the friction velocity $u^{*2}$. The magnitude of Reynolds stress increased with $y/h$ in the near bed region, reached a maximum around $y/h \approx 0.2$, and then decreased towards the free surface, for each scenario. An increase in Reynolds stress with increased bed mobility is evident. As plotted in Figure 6b, the data from Hanmaiahgari et al. [10], regarding incipient bed motion (IN) and mobile dunes (SM2), also show increased Reynolds stress with increasing mobility of bed forms. The data from Lichtneger et al. [11] (R1s) show a test run, with mobile sediments moving on an immobile bed.

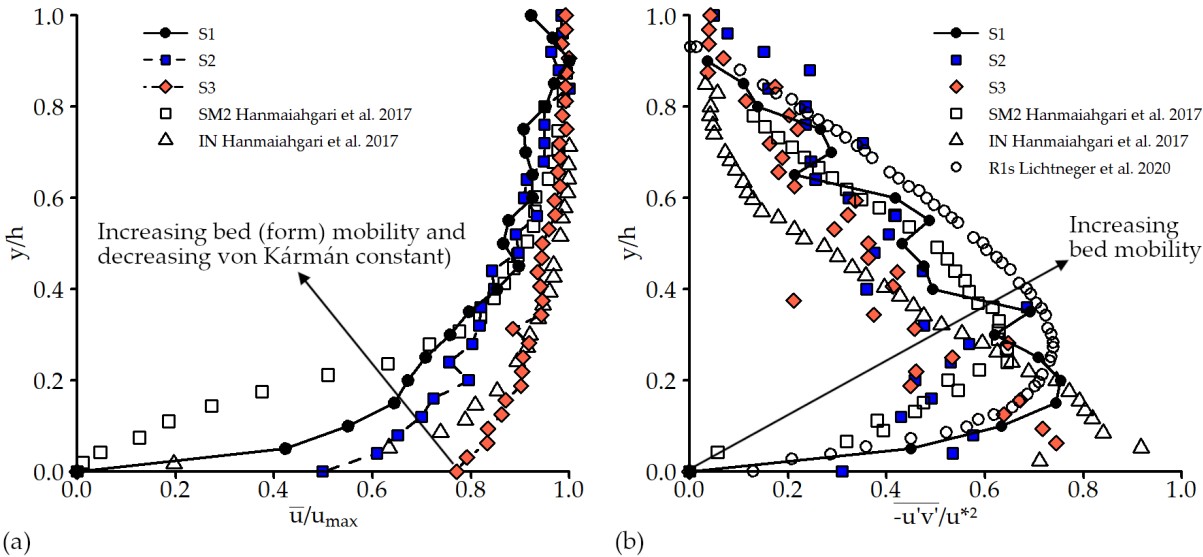

**Figure 6.** Results of the LDV measurements: (**a**) normalized mean velocity profiles; (**b**) normalized Reynolds stress profiles. In the data of Hanmaiahgari et al. [10], IN denotes incipient motion, and SM2 denotes mobile bed forms. In the data of Lichtneger et al. [11], R1s denotes a test run with additional sediment feeding and an immobile bed. The black arrows indicate increased sediment transport.

## 4. Discussion

Some key aspects, worth considering for future hydropower operations, are: (i) ecology; (ii) flood risk; (iii) technical issues and energy revenue [3]. Regarding ecology and river morphology, the conservation of sediment connectivity should be pursued [3]. This can be achieved if reservoirs are drawn down in periods of high sediment transport [3]. During projected operations, most of the bed load material is expected to settle in the headwater section (headwater delta formation), and is, therefore, able to locally increase

the flood risk [2]. From a technical point of view, intensive maintenance work or loss of hydraulic head caused by sedimentation should be avoided [3].

To discuss the interactions between gate position, flow dynamics, and sediment transport, we use the Shields diagram [14], which describes the initiation of sediment transport and sediment entrainment based on mean flow parameters. Shields [14] defines two dimensionless variables: (i) the Shields number $Fr^* = \tau_0/(\Delta g d_s)$, where $\Delta = (\rho_s/\rho) - 1 \approx 1.65$; and (ii) the particle Reynolds number $Re^*$. To be consistent with the cited literature [10,11,18–21], we now use the mean grain diameter $d_m$ for the calculation of $Re^* = u^*/(d_m\nu)$. If $Fr^*$, in dependence of $Re^*$, is larger than the critical Shields number $Fr^*_{crit}$ (threshold conditions of sediment entrainment), the Shields diagram predicts occurring bed forms or morphodynamics, such as dunes or riffles [20]. Figure 7 shows the Shields curve, together with the present results and the findings of others [10,11,18–21].

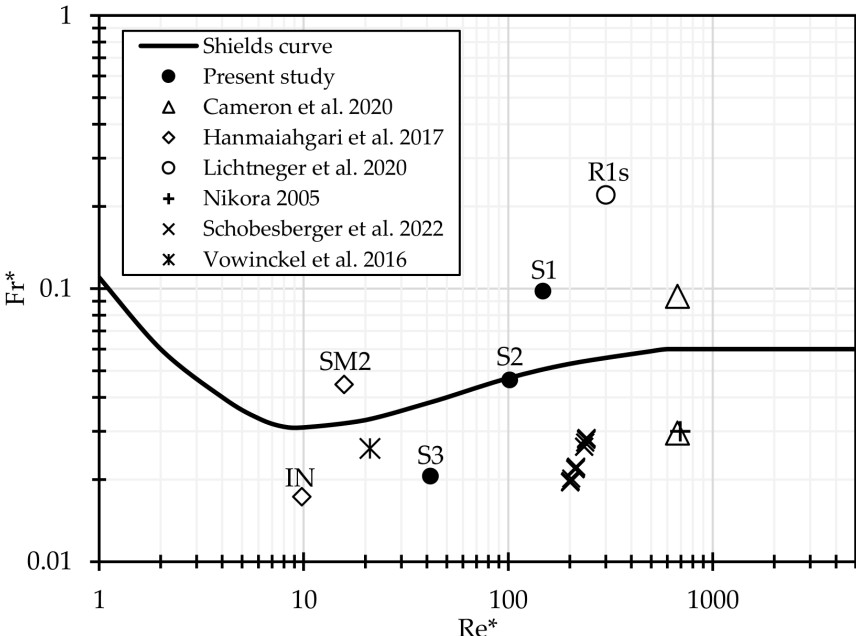

**Figure 7.** Shields diagram. In the data of Hanmaiahgari et al. [10], IN denotes incipient motion, and SM2 denotes mobile bed forms. In the data of Lichtneger et al. [11], R1s denotes a test run with sediment movement past an immobile rough bed. In addition, experimental data of Cameron et al. [18], Schobesberger et al. [20], field data of Nikora [19], and numerical data of Vowinckel et al. [21] are shown.

While S1 is well above the Shields curve in the region accompanied by sediment transport as dunes, S2 is precisely on the Shields curve, and S3 is located below critical conditions. This is also reflected by the measured sediment entrainment rates. We found intense bed mobility for S1, little bed load transport for S2, and no sustainably lasting sediment entrainment for S3. Hence, in each scenario, the Shields diagram and the experimental observations are in line. The initially high sediment output for S3 was only related to the locally occurring 3D erosion process (flushing cone in the vicinity of the weir).

The cited studies dealing with sediment entrainment below the critical Shields number mostly focused on turbulence and coherent structures [19–21]. Furthermore, Cameron et al. [18] reported higher drag forces during the passage of very-large-scale coherent motions (VLSMs). The findings of Cameron et al. [18], Nikora [19], Schobesberger et al. [20],Vowinckel et al. [21] help us to better understand the physics during single particle entrainment and incipient motion processes. This is crucial for the future development of physical-based models not reliant on mean flow parameters. Nevertheless, for the present work and a scaled model arrangement, the Shields curve serves it purpose well. Most closely comparable

to our conditions are the findings of Hanmaiahgari et al. [10], which show below- and above-critical Shields conditions—however, with lower $Re^*$.

Looking at our findings, together with the data of Lichtneger et al. [11] and Hanmaiahgari et al. [10], strong correlation between the Shields diagram and the Reynolds stress profiles is evident. Increase in $Fr^*$ and/or $-\overline{u'v'}$ correlates with increasing sediment transport and bed form mobility. As sediment transport and the formation of dunes require energy coming from the flow, there is another link to the available kinetic energy represented by the flow velocity. This manifests in the shift in the velocity profile (compared to the logarithmic law) and in the decrease in $du/dy$.

From a hydraulic perspective, our present findings further support the initiation of flushing operations at flow rates of about $0.7 \times HQ_1$, as suggested by Reiterer et al. [3] and Sindelar et al. [2]. For the investigated LFFFO, the fully opened gate arrangement (S1) led to sufficiently large Reynolds stresses to preserve bed mobility and to route large parts of the incoming sediment downstream. For flow rates exceeding this threshold value, even higher flow velocities and corresponding Reynolds stresses can be expected.

Considering ecology, LFFFO on a regular basis should be preferred, as higher flow rates not only are accompanied by higher bed load transport, but also lead to increasing suspended matter concentrations (turbidity), mobilized organic material, and oxygen consumption. While the gate operations investigated during the present study may not have had an ecological effect during the event, they possibly increased the amount of sediment routed downstream of the HPP which, in turn, improved the sediment connectivity.

From a flood risk perspective, the implementation of optimized gate opening sequences can also be beneficial. Sediment depositions upstream of the HPP (especially at the headwater section of the impoundment) can significantly decrease the effective flow area during extreme flood events, which, in turn, increases the flood risk [22]. To avoid such depositions, Harb et al. [22] and Reiterer et al. [3] suggest reservoir flushing on a regular basis at moderate flood flows (LFFFO), rather than infrequently at high flood rates, to continuously route incoming sediments downstream of the HPP. Therefore, implementing optimized gate position sequences during these events could further improve flushing efficiency, reducing depositions upstream of the HPP, and thus, the flood risk.

Considering the economic implications of the present findings, asymmetrical gate positions in the initial phase of a flushing event can be beneficial, for locally removing depositions in the vicinity of the weir. According to Harb et al. [22], depositions near the turbine intake should be avoided, to reduce maintenance costs and to prevent possible technical issues.

While in the present study scenario S1 featured Shields numbers $Fr^*$ well above the critical Shields curve, S2 and S3 lay on or even below the curve: this indicates the importance of full drawdowns, complementing the existing literature [3,4]. In addition to the significant effects of gate positions on flushing efficiency, Sindelar et al. [6] also underline the importance of low fixed weir heights and reduced reservoir widenings, to improve sediment continuity. Aside from management measures and the structural design of HPPs, other factors can also have a significant effect on flushing efficiency. Mao [23], for example, reports on the effect of different hydrographs on bed load transport rates. His findings indicate higher transport rates during rising limbs than during falling limbs. Based on this information, possible limitations of the present study are that the experimental setup and conditions did not account for different impoundment geometries, the HPP's structural design (widening and fixed weir height), and the impact of varying hydrographs on the bed load transport. Hence, these factors should also be considered when planning upcoming laboratory research and in the implementation of future sediment management strategies.

## 5. Conclusions

This paper dealt with the effects of gate opening positions on the flushing efficiency at RoR-HPPs during LFFFO. Based on an idealized physical model setting, consisting of two movable radial gates above a fixed weir, experiments were conducted in a hy-

draulic flume. Additionally, the hydrodynamic properties of the flow, and morphodynamic alterations, were recorded, using photogrammetry and advanced flow measurement techniques (2D LDV).

For the investigated gate positions, the full opening of both radial gates led to the highest flushing efficiency. Even for slightly submerged gates, the flushing efficiency rapidly decreased. This points out the importance of a full reservoir drawdown, to successfully route the sediments downstream of the HPP. For the asymmetrical gate opening (one gate closed, one gate open), a flushing cone formed in the vicinity of the weir. The 3D erosion process led to an initially high flushing efficiency, which rapidly decreased after a short period of time.

The different gate positions significantly influenced the hydrodynamic flow properties. With both gates fully open, and free flowing conditions, strong bed load movement, in the form of dunes, was present. The mobile bed forms and the high sediment transport led to a shift in the velocity profile, showing a decrease in $du/dy$, which was accompanied by increased normalized Reynolds stresses.

The present results could affect future HPP operations and sediment management strategies from a hydraulic perspective, by stressing the importance of full gate opening, to successfully route sediments through RoR-HPP impounded river sections. Eventually, taking the findings of the present study into consideration, a detailed investigation of different gate position sequences should be considered in upcoming research. A combined approach, starting the flushing phase with asymmetrical gate opening positions, followed by a full drawdown, could be a promising approach to further increase flushing efficiency. Additionally, the influence of varying hydrographs [23], different reservoir widenings, and fixed weir heights [6] on the respective Reynolds stresses and on bed mobility may be another good starting point for upcoming laboratory research.

**Author Contributions:** Conceptualization, T.G.; methodology, T.G. and K.R.; software, T.G. and K.R.; formal analysis, T.G. and K.R.; investigation, T.G. and K.R.; writing—original draft preparation, T.G. and K.R.; writing—review and editing, C.S., C.H., and H.H.; funding acquisition, C.H. and H.H. All authors have read and agreed to the published version of the manuscript.

**Funding:** The financial support provided by the Austrian Federal Ministry for Digital and Economic Affairs and by the National Foundation of Research, Technology and Development of Austria is gratefully acknowledged. We also gratefully acknowledge financial support from the Christian Doppler Research Association.

**Data Availability Statement:** The datasets analyzed during the current study can be made available by the corresponding author on reasonable request.

**Acknowledgments:** We thank bachelor students Eva Manzenreiter and Paul Georg Wiltsche for their assistance during the experiments.

**Conflicts of Interest:** The authors declare no conflict of interest. The funders had no role in the design of the study, in the collection, analyses, or interpretation of data, in the writing of the manuscript, or in the decision to publish the results.

## Abbreviations

The following abbreviations were used in this manuscript:

| | |
|---|---|
| DEM | digital elevation model |
| HPP | hydropower plant |
| $HQ_1$ | annual flood flow |
| LDV | laser doppler velocimetry |
| LFFFO | low-flood-flow flushing operations |
| PG | point gauge |
| PHOTO | camera system for photogrammetry |
| Q | flow rate |

| | |
|---|---|
| RoR | run-of-river |
| SIC | constant sediment input |
| ST | sediment trap |
| WM | weir model |

**Notations**

The following notations were used in this manuscript:

| | |
|---|---|
| $A_r$ | areal scale |
| $d_s$ | maximum grain diameter |
| $d_m$ | mean grain diameter |
| $Fr^*$ | Shields number |
| $Fr$ | Froude number |
| $h$ | water depth |
| $L_r$ | geometric length scale |
| $\kappa$ | Von Kármán constant |
| $\nu$ | kinematic viscosity |
| $Q_r$ | scale ratio for flow rate |
| $R_{ij}$ | Reynolds stress |
| $Re^*$ | particle Reynolds number |
| $\rho$ | water density |
| $\rho_s$ | sediment density |
| $t_r$ | hydraulic timescale |
| $t_s$ | sediment timescale |
| $\tau_0$ | reference shear stress |
| $u$ | measured longitudinal velocity |
| $\overline{u}$ | time-averaged longitudinal velocity |
| $u'$ | turbulent longitudinal velocity fluctuation |
| $u_{max}$ | maximum longitudinal velocity |
| $u^*$ | friction velocity |
| $v$ | measured vertical velocity |
| $v_r$ | velocity scale |
| $\overline{v}$ | time-averaged vertical velocity |
| $v'$ | turbulent vertical velocity fluctuation |

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
