# Peer review of "Flushing Efficiency of Run-of-River Hydropower Plants: Novel Approaches Based on Physical Laboratory Experiments"

_water, doi:10.3390/w15142657_

Round 1
Reviewer 1 Report
What is the scientific conclusion for the study? I think it would be better if its scientific contribution is explained in the abstract by a sentence (abstract, line 18)
I suggest to start the introduction section by definition of run-of-river hydropower that means a facility, which has channels flowing water from a river through a canal or penstock to spin a turbine and possessing no storage capacity. It is not a good way to start to the introduction section with the sediment continuity of many Alpine Rivers (section 1, line 20).
There is a comparison between findings of this study and the studies done previously on Discussion section (section 4, lines 235-238). It is not clear for new researchers. I recommend the fact that it should be clearer and more understandable for young engineers.
The authors should explained how to use the findings of study for forthcoming studies. They have mentioned to the scientific contribution of this study. The suggestion is briefly given in last paragraph for the forthcoming researches. However, it should be more understandable and include for new starting steps.
Reviewer 3 Report
Comments:
- What is the significance of periodic flushing operations during moderate river flood events?
- How do run-of-river hydropower plants (RoR-HPPs) contribute to disturbed sediment continuity in rivers?
- What factors are considered when determining discharge values for initiating gate operations during flushing measures?
- How does a discharge value of 0.7×HQ1 serve as a good reference point for initiating gate operations?
- What were the objectives of the physical model tests conducted in this study?
- How was the idealized model arrangement scaled down, and what sediment mixture was used?
- What methods were employed to measure bedload rates and 2D velocities near the moveable radial gates?
- What information was derived from the mean velocity profiles and Reynolds stresses obtained through Laser Doppler Velocimetry (LDV)?
- What are the effects of fully open non-submerged gates on bed load mobility?
- How does gate submergence affect bed load rates during the flushing operation?
- What are the implications of asymmetrical gate positions on the flushing process and the removal of depositions?
- Can you provide more details about the flushing cone created by asymmetrical gate positions?
- How effective are the different gate operation patterns in improving the flushing efficiency for RoR-HPPs?
- What are the potential ecological benefits of implementing effective gate operation patterns during flushing measures?
- How do these different gate operation patterns impact flood risk management in rivers?
- Are there any technical challenges or considerations associated with implementing the identified gate operation patterns?
- What are the economic implications of the different gate operation patterns for RoR-HPPs?
- How do the findings of this study contribute to the existing knowledge about sediment management in rivers affected by hydropower plants?
- Are there any limitations or potential sources of error in the physical model tests conducted in this study?
- What are the possible future research directions or practical applications based on the conclusions drawn from this investigation?
Minor editing of English language required.
Round 2
Reviewer 3 Report
The authors have addressed my comments; I suggest this manuscript can be accepted in its present form.